# Kids, Difficult Asthma and Fungus

**DOI:** 10.3390/jof6020055

**Published:** 2020-04-27

**Authors:** Andrew Bush

**Affiliations:** 1Department of Paediatric Respiratory Medicine, Royal Brompton Hospital Harefield NHS Foundation Trust, Sydney Street, London SW3 6NP, UK; a.bush@imperial.ac.uk; Tel.: +44-207-351-8232; Fax: +44-207-351-8763; 2Paediatric Respiratory Medicine, National Heart and Lung Institute, Imperial College, Sydney Street, London SW3 6NP, UK

**Keywords:** atopy, aspergillus bronchitis, fungal sensitization, itraconazole, severe asthma, voriconazole

## Abstract

Fungi have many potential roles in paediatric asthma, predominantly by being a source of allergens (severe asthma with fungal sensitization, SAFS), and also directly damaging the epithelial barrier and underlying tissue by releasing proteolytic enzymes (fungal bronchitis). The umbrella term ‘fungal asthma’ is proposed for these manifestations. Allergic bronchopulmonary aspergillosis (ABPA) is not a feature of childhood asthma, for unclear reasons. Diagnostic criteria for SAFS are based on sensitivity to fungal allergen(s) demonstrated either by skin prick test or specific IgE. In children, there are no exclusion criteria on total IgE levels or IgG precipitins because of the rarity of ABPA. Diagnostic criteria for fungal bronchitis are much less well established. Data in adults and children suggest SAFS is associated with worse asthma control and greater susceptibility to asthma attacks than non-sensitized patients. The data on whether anti-fungal therapy is beneficial are conflicting. The pathophysiology of SAFS is unclear, but the epithelial alarmin interleukin-33 is implicated. However, whether individual fungi have different pathobiologies is unclear. There are many unanswered questions needing further research, including how fungi interact with other allergens, bacteria, and viruses, and what optimal therapy should be, including whether anti-neutrophilic strategies, such as macrolides, should be used. Considerable further research is needed to unravel the complex roles of different fungi in severe asthma.

## 1. Introduction

The important role of fungi in worsening asthma has long been appreciated. In 2006, the term ‘severe asthma with fungal sensitization (SAFS)’ was first proposed in a review article that rightly acknowledged the historical evidence implicating fungi in the pathophysiology of asthma going back to the seventeenth century [1]. The role of fungi in asthma remains controversial to the present day, and these issues are reviewed below. What is certainly beyond dispute is that (a) although most acute attacks of asthma are precipitated by a viral infection, a sudden heavy aeroallergen load, such as grass pollen (“thunderstorm asthma”) [2] or soya bean (ships unloading in the docks of Barcelona) [3], can precipitate severe attacks, which might be eosinophilic rather than neutrophilic [4]; and, (b) fungal allergens can also cause acute attacks of asthma [5,6,7]. 

The term SAFS focuses on allergic sensitization, but this is quite restrictive, because allergy is not the only mechanism whereby fungi can modulate asthma. Additional to allergic sensitization, which does not necessarily require airway fungal infection, is the release of tissue damaging proteases and other enzymes, which might disrupt the airway epithelial barrier and cause mucosal damage and airway remodeling [8]. For this to happen, a chronic fungal bronchitis needs to be established. Sensitization and tissue damage both may co-exist. Here, I propose that the more general term ‘fungal asthma’ is used to encompass allergic sensitization (SAFS), fungal bronchitis, and combined sensitization/bronchitis (Figure 1). In adults, this would also include ABPA, not discussed here because of the rarity of this condition in children with asthma. All three entities may potentially benefit from anti-fungals, but fungal bronchitis without sensitization should not require the intensification of anti-Type 2 inflammatory medications. Of course, the pro-inflammatory effects of tissue damaging enzymes may merit treatment (as, for example, the anti-inflammatory strategies that may be used in cystic fibrosis (CF) [9,10] to counter the effects of infection driven, neutrophilic tissue destruction. The picture might also be dynamic; increasing inhaled steroids may cause topical immunosuppression (discussed in more detail below) and, thus, predispose to fungal bronchitis as a secondary phenomenon.

The justification for this sort of phenotyping is that it is clinically useful, because defining it leads to a change in management. Unfortunately, much of the paediatric guidance has had to be extrapolated from work in adults. The aim of this review is to assess the clinical utility of current concepts of fungal asthma (as defined above) in children, and suggest new approaches and where future work is needed. Although this review will focus as far as possible on children, it inevitably has to supplement this with adult experience and animal and cellular models where paediatric data are not available. Prior to writing this manuscript, a literature search was performed while using the search term <severe asthma with fungal sensitization> limited to English Language papers, which was supplemented from the author’s personal archive of references.

## 2. Definition of SAFS and Fungal Bronchitis

### 2.1. SAFS in Adults

SAFS was first defined in adults [11], and it has been suggested that it is a more severe phenotype than seen in unsensitized patients. For the purposes of the definition of SAFS, severe asthma is defined as treatment with 500 mcg Fluticasone/day or equivalent, or continuous oral corticosteroids, or four prednisolone bursts in the previous 12 months or six in the previous two years (as with so many definitions of severity, the figures are fairly arbitrary). The immunological criteria for SAFS in adults also include a total immunoglobulin (Ig)-E < 1000, and negative IgG precipitins to *Aspergillus fumigatus* (*AF*) because allergic bronchopulmonary aspergillosis (ABPA) is a diagnostic consideration in adults, and in order to differentiate between ABPA and SAFS. Additionally, there needs to be evidence of sensitization (skin prick test wheal (SPT) ≥ 3 mm, specific IgE (sIgE) ≥ 0.4) to at least one of seven fungi, namely *AF, Cladosporium herbarum, Penicillium chrysogenum* (*notatum*), *Candida albicans, Trichophyton mentagrophytes, Alternaria alternate,* and *Botrytis cinerea*. The question as to whether sensitization is best determined by sIgE or SPTs was addressed in 121 patients with severe asthma (British Thoracic Society/SIGN steps 4 and 5) who underwent both tests to all the above fungi, except *Trichophyton mentagrophytes* [12]. Fungal sensitivity was very common, but concordance between skin prick tests and sIgE tests was poor (77% overall, but only 14–56% for individual fungi). Hence, both of the tests need to be undertaken to rigorously diagnose SAFS.

### 2.2. SAFS in Children 

There is no consensus definition of SAFS in children. Empirically, we define SAFS as severe, therapy resistant asthma [13] (STRA, with any pattern of symptoms), and we have used the same sensitization criteria as in adults, although in fact in a clinical setting we usually can only test for *AF*, *Cladosporium* and *Alternaria alternate*. For reasons that are unclear, ABPA is rarely, if ever, seen in children with asthma, despite being relatively common in children with CF [14], and so we do not adopt the IgE and IgG precipitin criteria of the adult definition. It is likely, but unproven, that there will also be discordance between sIgE and SPT results in children also [15], so both tests are needed.

Table 1 contrasts the diagnosis of SAFS in adults and children.

### 2.3. Beyond SAFS: Fungal Detection in the Airway, Fungal Bronchitis and Asthma 

There is no requirement to detect fungi within the airway in order to diagnose SAFS, although fungal infection might be part of the syndrome. However low-grade fungal infection might drive asthma without inducing sensitization, for example, by the release of tissue damaging enzymes disrupting epithelial barrier function (below). In CF, *AF* bronchitis is associated with worse outcomes [16,17,18], giving biological plausibility to this mechanism in asthma. The isolation of fungus from airways of SAFS patients is unsurprisingly very common. *AF* sputum positivity by PCR was 70% in SAFS patients not taking anti-fungals [19], but the frequency was reduced in those prescribed these medications, and in a small subgroup in whom serial samples were obtained, itraconazole therapy resulted in sputum reverting from a positive to negative PCR. The sensitization to multiple molds is also common in asthma. In one study, 60% of patients were poly- sensitized, most frequently to *Aspergillus fumigatus* (32%) and *A. Alternata (28%), Penicillium chrysogenum, Penicillium brevicompactum, Cladosporium cladosporioides*, and *Cladosporium sphaerospermum* [20,21]

There is also the issue of how intensively the presence of fungi should be sought. CF definitions are largely based on positive cultures, although whether repeated cultures or a single culture is needed for diagnosis is controversial. Much of the focus has been on *AF*, not least because it grows at 37 degrees (body temperature) and the spores are aerodynamically well suited to lodging in the lower respiratory tract, but as already stated, many other fungi may be important. In a study in which 69 adults underwent FOB and BAL, no fewer than 86% had fungi detectable by PCR on BAL, 46% of which were *AF*. Although a positive BAL was associated with increased BAL and plasma cytokines, there was no relation to asthma severity [22]. Molecular techniques may be even more sensitive. This study suggests that, the harder fungi are sought, the more they will be found. This group reported no increased asthma severity in SAFS adults; and importantly, potentially broadened the spectrum of fungi to which the patient may be sensitized.

### 2.4. Fungal Asthma or Fungal Asthmas? 

It should be noted that the danger of umbrella definitions is that it could be taken to assume that all fungi have equal effects. The magnitude of the effects might be different, and will likely also be dependent on levels of exposure, but, more importantly, the pathophysiological pathways may be different. Clearly, if the approach is treatment with anti-fungals, this is irrelevant, but any molecular therapies may need to be fungus-specific (below).

## 3. Paediatric and Adult Severe Asthma and the Atopies: Important Differences Relevant to Fungal Asthma

The vast majority of children with severe asthma are markedly atopic [23], with multiple sensitizations to aeroallergens, such as house dust mite, grass and tree pollens, cockroach, and furry pets. By contrast, much severe adult asthma is neutrophilic, often in the obese and with other co-morbidities, and with a female preponderance [24,25]. It is also increasingly being realized that atopy is not ‘all-or-none‘ and can be quantified [26]. Different atopies have differing significances [27,28,29]. Furthermore, complex interactions between allergens may be more important than individual results [30]. Sensitization to fungi is one part of the atopies; the question is, whether there is a discrete entity of SAFS in children, or whether fungal sensitization is one facet of asthma with polysensitzation to aeroallergens; to some extent, this remains unresolved. It might also be that the significance of fungal sensitization will be different in adults, and more likely to be a discrete entitity rather than mark of multiple sensitization, and this needs further exploration. However, the issue of anti-fungal treatment for paediatric SAFS is more one of ‘does it work?‘ rather than ‘should it work?‘. 

## 4. Epidemiological Data: Associations between Fungi and Asthma Severity

### 4.1. Cross-Sectional Studies 

Most of the big studies are in adults. The European Community Respiratory Health Survey [31] studied 1132 adults aged 20–44 years with current asthma. The frequency of mold sensitization (*Alternaria alternata* or *Cladosporium herbarum*, or both) increased significantly with increasing asthma severity across Europe, but there was no association between asthma severity and sensitization to pollens or cats. However, *Dermatophagoides pteronyssinus* sensitization was also positively associated with asthma severity. Thus, mold sensitization was highly associated with severe asthma in adults, but not uniquely so. In a systematic review and meta-analysis of 20 studies from 13 African countries [32] the mean asthma prevalence was 6%. The prevalence of fungal sensitization, mostly on skin prick testing, ranged from 3% to 52%, mean 28% with a pooled estimate of 23.3%. *Aspergillus* species were commonest. The prevalence of ABPA was estimated at 1.6–21.2%. A similar study related fungal allergy to asthma severity, and there were no paediatric data. 

Another such study in severe asthma (GINA step 4 or 5 treatment) [33] enrolled 124 patients. A variety of markers were collected, including spirometry, exhaled nitric oxide, serum cytokines, and IgE. Fungal sensitization was assessed from IgE specific to fungal allergens (*AF*, *Alternaria*, *Candida*, *Cladosporium*, *Penicillium*, and *Trichophyton* species and the *Schizophyllum* commune). Thirty-six of 124 patients (29%) were sensitized to at least one fungal allergen, most commonly *Candida* (16%), *AF* (11%), and *Trichophyton* (11%). Early-onset asthma (<16 years of age) was more common in patients with fungal sensitization (45% vs 25%; *p* = 0.02, see below). Interleukin-33 levels were also higher in patients with fungal sensitization, as discussed in more detail in the sections on pathophysiology. Asthma Control Test scores were worse in patients with multiple when compared with single fungal sensitizations and non-sensitized controls. 

### 4.2. SAFS and Control of Asthma 

Adult SAFS patients are more likely to have uncontrolled symptoms [34,35,36,37,38,39]. In a retrospective review of urban adult asthma patients, total serum IgE was highest in the 53 patients (17.3%) with fungal sensitization (median, 825 IU/mL vs. 42 non-atopic (n = 137, 44%) vs. 203 other allergen sensitized (n = 117, 38.1%), *p* < 0.001). The fungal sensitized patients were more likely to have been admitted to the intensive care unit (ICU) admission and been ventilated (13.2% vs. 3.7% non-atopic vs. 3.4% other sensitization *p* = 0.02; and 11.3%, 1.5%, and 0.9%, respectively, for ventilation, *p* < 0.001). There are two possible interpretations of these data; firstly, polysensitized atopic asthmatics do worse, or that fungal sensitization is a discrete entity and an independent risk factor for bad outcomes.

A study [40], which evaluated 206 adults with severe asthma (GINA step 4 or 5 treatment, mean age 45 ± 17 years, 99 [48%] male), of whom 78% had a positive SPT to one or more allergens. The most common allergen reported was house dust mites (*Blomia tropicalis, Dermatophagoides pteronyssinus* and *Dermatophagoides farinii*), but 11.7% were sensitized to *Aspergillus* species, and this was associated with uncontrolled asthma. In particular, *Aspergillus* sensitization was independently associated with the need for ≥2 steroid bursts in the past year (odds ratio 3.05, 95% confidence interval 1.04–8.95). There was no association between asthma control and corticosteroid bursts with sensitization to any other allergen. Importantly, this study suggests that all fungi do not necessarily have equivalent effects.

### 4.3. SAFS and Asthma Attacks: Children 

Paediatric data are much scantier, but the conclusions are very similar. A German group reviewed 207 children with a diagnosis of asthma of varying severity (25% had mild, 31% moderate, and 44% severe; 26% had a previous history of hospitalization for an asthma attack [35]). *Alternaria* was the leading mold causing sensitization, but this did not correlate with hospitalization due to asthma attacks or other parameters of asthma severity. The prevalence of *Alternaria* sensitization increased with age and there was a significant association with the sensitization to other molds and aeroallergens, grass pollen, and cat epithelia. *Alternaria* sensitization in this study was thus not a risk factor for severe asthma and hospitalization. However, it should be noted that the risk might be a composite, both of sensitization, but also level of exposure; to take an absurd example, a sensitized patient who never subsequently encountered the allergen could not have an asthma attack triggered by that allergen.

The Melbourne Air Pollen Children and Adolescent study [41] recruited 644 children and adolescents (aged 2–17 years) that were hospitalized for asthma and showed that exposure to *Alternaria*, less well known taxa, including *Leptosphaeria*, *Coprinus,* and *Drechslera*, and total spore counts were significantly associated with admissions for asthma independent of rhinovirus infection. Surges of spores of *Alternaria*, *Leptosphaeria*, *Cladosporium*, *Sporormiella*, *Coprinus*, and *Drechslera* were associated with significant effects delayed for up to three days, and *Cladosporium* sensitization was associated with significantly greater effects than the other fungi. Importantly, this study broadens the range of fungi that might need to be considered as part of fungal asthma, although the alternative explanations are that the effects were not mediated by allergic sensitization, or less likely, that that these spores were merely possibly markers of some unidentified root cause.

### 4.4. SAFS and Lung Tissue Destruction 

In a cross-sectional study [42], 329 (76.3%) of adult asthmatics were sensitized to at least one fungus and this was related to the development of lung destruction, as assessed by post-bronchodilator spirometry and computed tomographic (CT) scans. The sensitization to *AF* and/or *Penicillium chrysogenum* was associated with a lower first second forced expired volume (FEV_1_) when compared with those not sensitized, independent of atopic status, and an increased frequency of CT abnormalities, bronchiectasis, tree-in-bud, and collapse/consolidation. Cluster analysis identified three clusters: (i) hypereosinophilic hypothetically, true SAFS; (ii) high immunological biomarker load and high frequency of radiological abnormalities (hypothetically, fungal bronchitis dominant; and, (iii) low levels of fungal biomarkers (fungi not relevant). The authors concluded that *AF* sIgE was a risk factor for lung damage irrespective of ABPA.

### 4.5. Fungi and Risk Assessment 

GINA and other guidelines have rightly stressed the importance of risk assessment as well as asthma control. There is no question that fungal sensitization is a marker of future risk of poor control and asthma attacks. Whether this is true for fungal bronchitis, as it is in CF, has yet to be explored.

## 5. Clinical Features of SAFS in Children

We have reported the largest, most detailed series of children with SAFS [43]. We studied 82 children (median 11.7 years) with severe, therapy resistant asthma (STRA), who had undergone a protocolised series of investigations [44,45,46], including fibreoptic bronchoscopy (FOB), bronchoalveolar lavage (BAL), and endobronchial biopsy (EBx). Thirty eight were defined as SAFS, with a specific IgE or SPT to *AF*, *Alternaria alternate* or *Cladosporium* (in practice, we do not have access to testing for other fungi). We also found that children with SAFS had an earlier onset of symptoms (0.5 as compared with 1.5 years), a higher IgE (637 vs. 177) and were sensitized on testing sIgE to more non-fungal inhalant allergens, when compared with non-SAFS STRA. They were more likely to be prescribed maintenance oral corticosteroids (42% vs. 14%, *p* = 0.02). However, on BAL and EBx, the severity of airway inflammation and remodelling (absolute thickness of reticular basement membrane thickness and airway smooth muscle mass) did not differ between SAFS and control STRA, despite the greater use of anti-inflammatory medications. Eight of 10 (80%) SAFS children responded to omalizumab, similar to STRA controls (11/18, 84%, *p* = NS). Mepolizumab was not licensed in children at the time of this study.

Another paediatric study enrolled 64 children, of whom 25 (39%) had evidence of sensitization to at least one fungus [47]. Nineteen of 25 (76%) sensitized children had severe persistent asthma when compared to 13 of 39 (33%) non-sensitized (*p* = 0.0014). Nineteen of 32 (59%) severe persistent asthmatics had fungal sensitization, and these also had higher serum IgE and worse spirometry. Bronchial biopsy of sensitized children revealed that these children exhibited basement membrane thickening and eosinophil infiltration on bronchial biopsy.

In a USA study of 126 children, *Alternaria* skin test reactivity was associated with severe, persistent asthma. Importantly, this was an independent risk factor to that of the total positive skin tests, suggesting there is an independent effect of this fungus unrelated to degree of atopy [48]. 

## 6. Treatment of SAFS and Fungal Asthma

The possible aspects of treatment are: (a) the reduction of allergic inflammation; (b) reduction of fungal burden; (c) reduction of tissue damage; and, (d) modulating the pro-inflammatory effects of tissue destruction. Most focus has been on the reduction of fungal burden, but without necessarily ensuring that there is a fungal infection. 

### 6.1. Adult Data 

Most of the data on antifungal therapy are in adults, and the results are conflicting. The FAST study [11] enrolled 58 adults into a double blind, randomized controlled trial of oral itraconazole or placebo for 32 weeks, with a follow up period of 16 weeks. The primary end point was the Asthma Quality of Life Questionnaire (AQLQ), with secondary endpoints being rhinitis score, total IgE and respiratory function. The study was positive, with improvements in AQLQ and rhinitis scores, an improved morning peak flow (20.8 l/min.) with itraconazole, and total IgE dropped (−510 iU itraconazole when compared with +30 placebo). Seven patients in the itraconazole group, and two placebo patients discontinued treatment. Interestingly, 60% had big improvements in QoL with itraconazole. The benefits of itraconazole declined rapidly in the washout period. By contrast, EVITA3 was a randomized, double blind, placebo controlled trial of Voriconazole in SAFS [49]. Of note, Voriconazole does not increase steroid bioavailability, unlike itraconazole [50,51]. The study duration was three months with a nine-month follow period. Fifty-six adults with SAFS were recruited. The inclusion criteria were at least two severe asthma attacks (defined as the prescription of oral corticosteroids) in the previous year, and a positive specific IgE or skin prick test to *AF*. The voriconazole levels were measured to optimize therapy. The primary endpoints were quality of life and asthma attacks. The study was negative. Neither trial mandated a positive airway fungal culture. In another report, 41 patients were studied retrospectively [52]. In those who received treatment (*n* = 32), this was with any of terbinafine, fluconazole, itraconazole, voriconazole, or posaconazole combined with standard treatment, by comparison with nine patients who had standard asthma therapy only. Those that were treated with anti-fungals showed improvement in Asthma Control Test, peak flow rate, and IgE. The response was better with longer treatment periods, and it was well tolerated, but relapse was common after the discontinuation of treatment. It should be noted that all of these data largely predate the widespread introduction of biological and, therefore, should be interpreted with caution in light of new therapies [53].

In another study [54], 110 STRA GINA stage 4 adult asthmatics were randomly assigned to 200 mg itraconazole twice a day or 10 mg prednisolone once daily for four months. There was no requirement for the demonstration of fungal sensitization or any other manifestation of fungal asthma. The study was not blinded. 71% of the itraconazole group improved and there were very few side-effects, whereas there was minimal change with prednisolone. 

In terms of acute asthma, there is a single case report of an 83 years old woman [55], with a 33-year history of asthma prescribed inhaled and oral corticosteroids. She presented with an acute attack of wheeze that did not respond to oral corticosteroids and antibiotics. She was found to culture *AF* in her sputum, a positive *AF* sIgE and IgG precipitins, and a positive galactomannan. Voriconazole was added with a good response. Perhaps the most likely explanation is that this was treating acute *AF* bronchitis in an immunosuppressed adult. There is no general role for antifungals in acute asthma.

### 6.2. Paediatric Data 

There are no randomized controlled trials of treatment in children. On general principles, we try to minimize fungal exposure, especially advocating for rehousing if there is visible mold in the house; we would check any nebulizers which might be being used for fungal contamination; and we would advise against children going into stables and barns [56], where mold abounds. However, although the reducing the burden of fungal allergen exposure seems sensible, the relationship between mold exposure, mold sensitization, and asthma severity is complex. Approximately 90% of homes in one case control study were contaminated with mold [20]. The sensitization to *AF*, but not to other molds, was associated with asthma severity. Whether or not the child was sensitized, *AF* and *Penicillium* spp. in dust was associated with severe asthma; the latter was associated with worse lung function. The lessons of this study are that environmental *AF* exposure should be minimized, and that not all molds have the same effects. However, although exposure to mold may limit airway infection, it should be borne in mind that allergen reduction strategies have sometimes had unexpected effects. In some cases, high level exposure might induce tolerance, and reduction in levels lead to increased sensitivity; and there is marked variation between allergens in the relationship between environmental concentrations and likelihood of sensitization or tolerance [57]. 

Additionally, we would also optimize standard asthma management and treatment [6,7,8], including treatment with omalizumab and mepolizumab if indicated, before going on to ‘beyond guidelines’ therapy with antifungals. Anecdotally, a child with refractory asthma, persistently abnormal spirometry, total E >20,000 IU/mL, and severe airway eosinophilia was sensitized to multiple fungi and responded dramatically to itraconazole [58]. Additionally, anecdotally, omalizumab might effectively treat the occasional case of SAFS [59], alone or combined with itraconazole which may be used to reduce IgE levels into the omalizumab range [60]. Also anecdotally, we have seen the occasional SAFS child who appeared to improve with antifungals.

### 6.3. Conclusions: What Is the Role of Antifungals in SAFS? 

It is suggested that the individual facets, or treatable traits of fungal asthma, are determined on an individual basis and a bespoke treatment plan developed. Clearly, the evidence for the use of antifungals is conflicting and of low quality. Part of the reason might be that SAFS as conventionally declined does not require the presence of fungal bronchitis. It is difficult to see how anti-fungal therapy would benefit SAFS if there were no fungal infection, and symptoms were solely due to sensitization to fungal spores. Logically, future trials of anti-fungal therapy in SAFS/fungal asthma should mandate the presence of fungal bronchitis.

Our current approach is to address environmental exposures and optimize standard therapy in children with SAFS. If asthma control is optimal and there are no other markers of ongoing risk, such as a persistently raised exhaled nitric oxide or a past history of really severe attacks, with no present side-effects, then we would not use antifungals. However, if asthma control remains suboptimal, or significant risks persist, then we would consider adding an antifungal, such as itraconazole. It is important to note that there is a potential interaction between corticosteroids and azoles at the cytochrome p450 level [61], such that the combination of itraconazole and inhaled budesonide has led to iatrogenic Cushing Syndrome [50,51].

## 7. Risk Factors for SAFS: Genetic Studies

Although there is expanding literature on the genetic associations of ABPA, SAFS has been little investigated. There might indeed be genetic factors that are associated with SAFS, but, to my knowledge, there has been no large scale Genome Wide Association Study (GWAS) to confirm or otherwise this suggestion. In a small, preliminary study [62], 325 haplotype-tagging single nucleotide polymorphisms (SNPs) in 22 previously suggested candidate genes were studied in SAFS (*n* = 47), atopic asthmatics (*n* = 152), and healthy control patients (*n* = 279). There were significant associations of Toll-like receptors (TLR) 3 and 9 (TLR3), C-type lectin domain family seven member A (dectin-1), IL-10, mannose-binding lectin (MBL2), CC-chemokine ligand 2 (CCL2) and CCL17, plasminogen, and the adenosine A2a receptor, different from those reported in asthma complicated by ABPA. Some of these hits are supported by cell and animal data (below). The main weakness of this study was the absence of a second validation cohort, without which the findings are, at best, preliminary. In an initial small study comprising 76 adults with chronic cavitatory pulmonary aspergillosis (*n* = 40), ABPA (*n* = 22), and SAFS (*n* = 14), no genetic associations of SAFS could be determined, unsurprisingly with such a small number of patients [63]. However, in one intriguing study, six SAFS children were heterozygous for a 24-base pair duplication in the CHIT1 gene [64]. This duplication associates with an increased susceptibility to fungal infection and decreased circulating chitotriosidase levels [65]. Clearly there is a need for more work in this area.

## 8. Pathophysiology of SAFS and Fungal Asthma

### 8.1. Introduction

As discussed, there are two pathological mechanisms, whereby fungi, especially *AF*, can cause disease in children with asthma [8]. These are as a source of allergen(s) to which the child is sensitized, leading to wheeze on exposure, and driving a Type 2 inflammatory response; and, the release of tissue damaging enzymes by fungi that have infected the airway (not dissimilar to, for example, house dust mite, which is allergenic and tissue damaging), and that might also generate an allergic response. It should be noted that other proteins could generate an allergic response without requiring airway infection, for example, in sensitization to furry pets.

Any account of the potential role of exogenous infection of any cause must consider the possibility that this is iatrogenic, secondary to the use of corticosteroids. It is known that systemic corticosteroids are immunosuppressive, and also that mucosal immunity is essential for normal host defence [65]. It is biologically plausible that topical steroids would be immunosuppressive, and indeed their use is associated with increased prevalence of tuberculosis, [66] atypical *Mycobacterial* infection [67], and, in patients with COPD, pneumonia [68]. It is virtually impossible to dissect out the contribution of inhaled corticosteroids (ICS) to SAFS, because, by definition, all SAFS patients will be prescribed ICS. In one study [69], the fungal microbiome (mycobiome) was determined on bronchoscopic samples. The investigators reported that the mycobiome was highly varied with the biggest load in severe asthmatics. Healthy controls had low fungal loads; the most common fungus detected was the poorly characterized *Malasezziales*. *AF* was most the common in fungus in asthmatics and accounted for the increased fungal burden. Corticosteroid treatment was significantly associated with an increased fungal load. These interesting data cannot unravel whether inhaled corticosteroids caused SAFS, or SAFS led to the prescription of more inhaled corticosteroids.

### 8.2. Cell and Animal Studies 

A number of different pathways have been implicated in SAFS, including the pattern recognition receptors (PRRs) TLR3, TLR9, and Dectin-1 and IL-7, Il-10, IL-22, CCL2, and CCL17 [70,71]. IL-33 has been implicated in both adult [36] and paediatric SAFS [5]. IL-33 is an epithelial alarmin, together with IL-25 and TSLP. It is a member of the eleven member IL-1 family of cytokines. Of these, seven are proinflammatory (IL-1α, IL-1β, IL-18, IL-33, IL-36α, IL-36β, and IL-36γ) and four probably immunomodulatory (IL-1 receptor antagonist [IL-1RA], IL-36Ra, IL-37, and IL-38). A recent manuscript [72] demonstrated that IL-1α and IL-1β are elevated in the BAL and sputum from adult SAFS patients. The same group used a murine model utilizing the *AF* challenge to show that IL-1R1 signaling promotes increased airway hyper-responsiveness and neutrophilic inflammation associated with type 1 (IFN-γ, CXCL9, CXCL10) and type 17 (IL-17A, IL-22) responses, each exacerbated in IL-1RA^−/−^ mice. The administration of human recombinant IL-1RA (Kineret/anakinra) abrogated these responses, all suggesting that IL-1R1 signaling via type 1 and type 17 responses is an important and potentially treatable pathway of SAFS.

A murine model further explored the links between *Alternaria* and asthma [73]. Wild-type and mice lacking the IL-33 receptor (ST2^−/−^) underwent inhalational challenge with inhaled house dust mite, cat dander, or *Alternaria*. Mice that were sensitized with house dust mite were subsequently challenged with *Alternaria* (with or without serine protease activity having been knocked down), and inflammation, remodeling, and lung function assessed 24 h after the challenge. Only *Alternaria* possessed intrinsic serine protease activity that led to the release of IL-33 into the airways via a mechanism that is dependent on the activation of protease activated receptor-2 and adenosine triphosphate signaling. This led to more pulmonary inflammation relative to that produced by the house dust mite challenge. IL-1β and matrix metalloproteinase (MMP) 9 release were also features of *Alternaria* challenge. Furthermore, *Alternaria* triggered a rapid, augmented inflammatory response, mucus hypersecretion, and airway obstruction. The effects of *Alternaria* were critically dependent on ST2 signaling. Hence, *Alternaria*-specific serine protease activity resulted in rapid IL-33 release, leading to TH2 inflammation and exacerbation of allergic airway disease.

Alternaria proteases may have an important role. One study [74] used cells from normals or patients with severe asthma. They used both 16HBE cells and fresh bronchial epithelial cells cultured to air-liquid interface (ALI), and challenged them apically with extracts of *Alternaria* in order to further explore the role of *Alternaria* proteases. *Alternaria* extract protease activity was determined and selective protease inhibition was studied. In 16HBE cells, *Alternaria* extracts stimulated release of IL-8 and TNFα from 16HBE cells and reduced trans-epithelial resistance (TER); serine proteases were the predominant effectors. The ALI cultures from asthmatic donors had a lesser IL-8 response to *Alternaria* when compared with non-asthmatic healthy controls. *Alternaria* only reduced TER in asthmatic cell cultures. Hence, severely asthmatic individuals may have bronchial epithelium that is more susceptible disrupted barrier function by *Alternaria*, thus potentially increasing susceptibility to other aeroallergens.

This finding might account for a report [75] showing that there might be significant interactions between fungal and other allergens, leading to a worsening of asthma. In a murine model, ryegrass sensitization and challenge unsurprisingly led to lung eosinophilia. *Alternaria* extract was given as a single challenge 3 days before or after challenges with ryegrass. This increased airway eosinophilia, peribronchial inflammation, and mucus production when compared with ryegrass-only challenges. Alternaria extract increased airway Type 2 innate lymphoid cells (ILC)2 and TH2 cell recruitment in ryegrass-challenged mice. If this translates into human asthma, this synergy could be another mechanism for the worse phenotype of SAFS/fungal asthma.

Another potentially relevant *Alternaria* protease pathway has been studied in a cellular model [76]. *Alternaria* induced the PAR_2_/β-arrestin-2 signaling pathway. Filtrates of *Alternaria* were used to sensitize and challenge wild-type, PAR_2_^−/−^ and β-arrestin-2^−/−^ mice. Intranasal challenge led to a protease-dependent upregulation of airway inflammation and mucin production only in wild-type mice. The administration of *Alternaria* alkaline serine protease (AASP), fully activated PAR_2_ signaling that induced eosinophil and lymphocyte recruitment, which was β-arrestin-dependent. 

The mice were nasally challenged with filtrates of wild type *AF*, and Asp f5 and Asp f13 null strains [77]. The null strains had equivalent effects on Type 2 inflammation and airway responsiveness. The protease null strains, especially the Asp f 13 knockout, resulted in significantly reduced airway inflammation and remodeling as compared with the wild type *AF* filtrate. *AF* might also be implicated in epithelial disruption. The serine protease Asp f13 and alkaline protease 1 (Alp 1) promote airway hyper-responsiveness by infiltrating the bronchial submucosa and disrupting the interactions between airway smooth muscle (ASM) and extracellular matrix (ECM) [78]. Alp 1-mediated ECM degradation leads to bronchoconstriction. This is further suggestive evidence that the disruption of the epithelial barrier might allow environmental allergens to access to the submucosa, thus promoting a cascade of adverse effects.

Hyaluron (HA), which is a major component of the extracellular matrix, might lead to the recruitment and activation of inflammatory cells. In a model of murine *AF* airways disease, the HA levels were elevated in allergic animals and the increase correlated with cellular inflammation cells [79]. The increase in HA levels appeared due to the upregulation of hyaluronidase-1 (HYAL1) and hyaluronidase-2 (HYAL2). There was co-localization of HA and new collagen synthesis and deposition, hence another mechanism whereby *AF* might contribute to airway remodelling. 

The endothelin pathway has also been proposed to have a role in SAFS [80]. *AF* increased epithelial endothelin-1 secretion by epithelial cells in vitro, but the other significant pro-fibrogenic factors TGF-β1, TGF-β2, and periostin were not affected. *AF* upregulated endothelin-1 in murine lungs, and this was associated with airway inflammation and remodeling, abrogated by the selective endothelin-1 receptor antagonist BQ-123.

### 8.3. Pathophysiology of SAFS in Adults 

Human asthmatics that were sensitized to fungi had raised levels of IL-7 in BAL, which was negatively correlated with airway responsiveness [70] This observation was taken forward in a murine experimental model of fungal asthma [74]. IL-7 administration worsened lung function and increased TH2 and pro-inflammatory cytokines (IL-4, IL-5, IL-13, and IL-1α, IL-1β, respectively) and pro-allergic chemokines (CCL17, CCL22) IL-7 administration also increased IL-22 from γδ T cells, iNKT cells, CD4 T cells, and ILC3s. IL22 is also implicated in the pathophysiology of experimental fungal asthma. IL-7 administration only increased iNKT cells in the presence of both type 2 and IL-22 responses. When IL-7Rα was blocked in an in vivo, there was reduction in IL-22 production, and also lower levels of CCL22, reduction in iNKT and CD4 T-cell numbers, and the recruitment of eosinophils. Dynamic lung resistance was increased, but mechanism was not clear. These results data implicate IL-7 as at least one modulator of SAFS. 

### 8.4. Pathophysiology of SAFS in Children 

We have focused in particular on the alarmin IL33. It has long been appreciated that the epithelium is not a mere passive barrier, a sort of biological cellophane, but it has a large suite of receptors, and it responds by signalling systemically when these receptors are activated. The classical epithelial cytoklines, in addition to IL33, are thymic stromal lymphopoietin (TSLP) and IL25. IL33 interacts directly with mast cells, and via ILCs with B cells, eosinophils, and T cells, driving the latter down a TH2 differentiation pathway [81]. IL-33 is part of the innate immune system and it is secreted by epithelial and smooth muscle cells. It has been implicated in causing airway inflammation in adults with asthma [82] and binds to the ST2 receptor. A GWAS also identified IL33 as being potentially important in asthma [83]. Our data in an otherwise unselected population of children with severe asthma showed that IL-33 appeared to be an important mediator [84]. We also showed that IL33 stimulated collagen secretion from cultured fibroblasts from paediatric EBx, and that this process is not responsive to corticosteroids. Of note, airway fibroblasts also express ST2, the IL-33 receptor. Hence, unlike the signature TH2 cytokines (IL4, IL5, IL13), IL33 appears to be steroid non-responsive, and we speculated that this cytokine could be a novel therapeutic target in STRA/SAFS. We extended our observations in the SAFS cohort that is described above [43], and additionally explored pathophysiology in an age-appropriate murine model [85]. We showed that there was increased IL-33 expression in BAL and endobronchial biopsies from children with SAFS when compared to those without fungal sensitisation, despite the former group being prescribed a higher level of treatment. We then used our established neonatal murine allergic airways disease model [85] to compare the effects of house dust mite antigen and *Alternaria*. We found that the mice developed the same increased levels of airway responsiveness as compared with placebo, but that total lung inflammation and airway eosinophilia were greater in the *Alternaria* exposed mice. Furthermore, although levels of IL4, IL5, and IL13 did not differ between the two challenges, there was more IL33, matrix metalloproteinase 9 (MMP9), IL13 positive ILCs, and T-cells in the *Alternaria* challenged mice. These effects were abrogated in mice that were null for the ST2 receptor. Furthermore, we showed that treatment with corticosteroids did not abrogate *Alternaria* induced airway responsiveness or BAL inflammatory changes, but the whole lung inflammatory parameters were partially reduced by steroid treatment. Additionally, corticosteroids reduced neither lung IL13, nor IL13 positive ILCs or T-cells in the presence of *Alternaria*. Accordingly, in summary, there are data implicate the epithelial alarmin IL33 in STRA and SAFS. It should be noted that all of these patients will have been prescribed high dose inhaled corticosteroids, which might have switched off the TH2 pathway. The effects of IL33 appear to be modulated through the ST2 receptor. In a murine model, SAFS, at least related to *Alternaria*, might be IL33 mediated and steroid resistant. These data suggest that novel strategies, such as an anti IL33 monoclonal or an ST2 antagonist, could be explored. The finding of elevated MMP9 in the *Alternaria* challenged mice is provocative. It suggests that we might need anti-neutrophil strategies as part of SAFS therapy. Additionally, we speculate that there might be neutrophilia and MMP-9 release in response to tissue damaging enzymes secreted by fungi in children given the rarity of neutrophilic asthma in our series [23]. Our murine data suggests that reducing the fungal burden might be beneficial because of the effects of *Alternaria* on corticosteroid responsiveness. 

## 9. Unsolved Problems and Future Work

There are more unsolved questions than answers arising from this literature review, as summarized in Table 2. The first is the question as to whether fungal asthma is a separate sub-phenotype, and not just a manifestation of poly-sensitized atopic asthma. The evidence is that fungal allergens and, in particular, the ability of fungi to infect the airway, means that fungal asthma is different, and the ability of antifungals to reduce airway infection in a specific subgroup of asthmatics offers a specific treatment. However, there might be developmental differences in the role and significance of fungi in asthma. Next, the question is whether the concept of SAFS being a subset of fungal asthma, in other words, that there is more to fungal asthma then just allergy. Again, the evidence from other airway diseases, in particular CF, is that it should at least be further explored.

Granting the above two premises, further issues remain. We cannot assume that all fungi cause airway disease by the same pathomechanisms and, if specific treatments beyond a generic antifungal are to be discovered, we need to be aware that pathways, especially those leading to tissue destruction, may be fungus specific. We also need more data on whether less commonly detected fungi can cause fungal asthma. Furthermore, fungi do not exist in isolation, and we need to know more about synergies with other allergens, perhaps as a result of fungal epithelial damage, and also viruses, which are known to synergize with allergen sensitization and exposure [86] The role of bacterial infection in causing asthma attacks has also come to the fore, and interactions with the microbiome are likely important. We need biomarkers to distinguish fungi which are of pathological significance from those which are harmless airway colonisers, especially since molecular techniques will likely detect fungi in everybody’s airway. The issue of when to give anti-fungal treatment needs careful reconsideration in the age of the new biologicals, and should perhaps be limited to patients with low grade fungal bronchitis. Finally, given the neutrophil recruitment and activation by fungus-induced tissue, damage consideration might be needed for anti-inflammatory strategies targeted at neutrophils, as well as standard type 2 inflammation [87].

## 10. Summary and Conclusions

I have proposed the umbrella concept of ‘fungal asthma’, comprising SAFS and fungal bronchitis, which over-lap but are different facets of how fungi can interact with the airways. Both are umbrella terms, and different fungi may exert their effects by different pathways. We need more data in children, but treatment should be optimizing the treatment of Type 2 inflammation for sensitization of fungi or their products, and anti-fungals for chronic infection. Whether modulating neutrophilic inflammation is indicated is another strategy that requires testing. The interactions between fungi, other aeroallergens and respiratory viruses and bacteria, also merit further study. We have much more to learn regarding the importance of fungi in asthma, both in adults and children.

## Figures and Tables

**Figure 1 jof-06-00055-f001:**
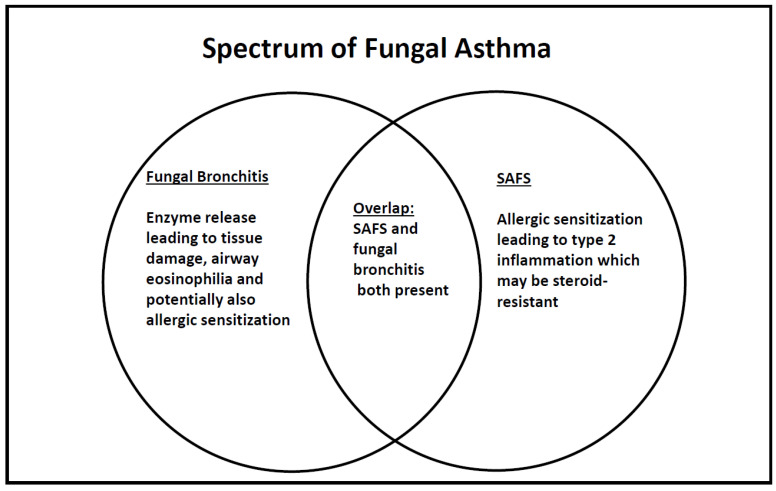
Schematic of fungal involvement in asthma in children, in whom the diagnosis of allergic bronchopulmonary aspergillosis (ABPA) is rarely made.

**Table 1 jof-06-00055-t001:** Diagnostic criteria for severe asthma with fungal sensitization (SAFS) in adults and children.

Fungal Sensitization (Positive Skin Prick Test and/or Specific IgE to One or More Fungus)	Other Adult Criteria	Other Paediatric Criteria
*Aspergillus fumigatus* *Cladosporium herbarum* *Penicillium chrysogenum* (*notatum*) *Candida albicansa* *Trichophyton mentagrophytes* *Alternaria alternate* *Botrytis cinere*	Treatment with 500 mcg Fluticasone Propionate/day, or Continuous oral corticosteroids, or 4 prednisolone bursts in 12 months or 6 bursts in 24 months	Severe, therapy resistant asthma (ERS/ATS Task Force criteria)
	IgE < 1000	IgE can be any level
	Negative IgG precipitins to *Aspergillus fumigatus*	IgG precipitins to *Aspergillus fumigatus* can be positive or negative

**Table 2 jof-06-00055-t002:** Unanswered questions in the field of fungal asthma.

No.	Unanswered Questions
1	Is the concept of fungal asthma, comprising SAFS and low-grade fungal bronchitis, a useful one?
2	Are fungal allergens qualitatively different in their effects from other aeroallergens, or is fungal sensitization merely a manifestation of poly-sensitization?
3	Is the significance of fungal asthma different in children with severe asthma, when multiple aeroallergen sensitization is much more common, compared with adults?
4	There are multiple fungi which could be significant, and molecular techniques will detect fungi with ever greater sensitivity, so we what biomarkers will enable us to differentiate fungi causing pathology from those which are harmless commensals?
5	Are there different fungal asthma, with different molecular pathways; in other words, are all fungi equal and equivalent, which seems unlikely?
6	How do fungi interact with other aeroallergens, viruses and bacteria within the airway?
7	Should anti-neutrophilic strategies such as azithromycin be used to mitigate the effects of neutrophilic inflammation and tissue damage?

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
