# Peer review of "Kids, Difficult Asthma and Fungus"

_jof, 2020, doi:10.3390/jof6020055_

Round 1

Reviewer 1 Report

Dr. Andrew Bush described the important issue regarding difficult to treat asthma and fungal sensitization.

The review was comprehensive with newly added references.
This manuscript will generate new ideas regarding the issue and will benefit the readers leading to better care for such patients.
The review is well written to publish after minor spelling miss and inconsistent abbreviation (eg IL-13).

Author Response

Thanks for your nice comments and the time you have spent on the manuscript. Both are much appreciated and I have done my best to identify and correct the spelling and abbreviations

Reviewer 2 Report

The review provides a very interesting overview on the current status and comparison of fungal related asthma in adults and children which is of interest to many research fields. The author raised many interesting questions to be addressed in different research fields and which can be of importance for future management of therapies. The manuscript is well written and has a clear story line.

My comments are mostly textually with a few questions to be addressed

Questions

Line 255 indicates that Ebx does see effects now (exhibited basement membrane thickening and eosinophil infiltration on bronchial biopsy), why is this not seen via Ebx earlier as mentioned in line 245/246? This requires some explanation.

Lines 430-433 this reads as a contradiction for Asp f13?  line 432-433 states that Asp f13 is less effective whereas l 431 indicates equivalent effects between WT and Asp F5 and f13 mutant strains? Please explain or adapt

Line 491 IL33 and its role, is this only in children? or also adults? Maybe the author can add such comment or explain the status

Textual

Lines

72/392/392/418/420 fungal names in italics; please check the complete manuscript

101-102/150-152  write names of all fungi used frequently as abbreviations or all in full length

132 rather

142 was 6% (add was)

147 start of the sentence is strange: remove such or write A similar study….

150 determining serum specific IgE specific to fungal: remove underlined specific

175-176 font sizes differ

187 been admitted to the intensive care unit (ICU) admission and.. : remove underlined word

195 Dermatophagoides  emode : there is regularly some extra space between the words

254 Rephrase : ..revealed that these children....

288 STRA and 371 ICS abbreviations were not yet introduced

343 GWAS abbreviation needs introduction here; later in line 470 it is used in full

438 to gain access to the ...

445/449 remodeling

452 reposition ref 74

465: cytokines

467/475  TH2 vs Th2

541 so which in stead of ,so we what…

Author Response

Thanks so much, please see the following responses:

The review provides a very interesting overview on the current status and comparison of fungal related asthma in adults and children which is of interest to many research fields. The author raised many interesting questions to be addressed in different research fields and which can be of importance for future management of therapies. The manuscript is well written and has a clear story line.

I appreciate these very nice comments, and the time taken to review the manuscript

My comments are mostly textually with a few questions to be addressed

Questions

Line 255 indicates that Ebx does see effects now (exhibited basement membrane thickening and eosinophil infiltration on bronchial biopsy), why is this not seen via Ebx earlier as mentioned in line 245/246? This requires some explanation.

THANK YOU. THE HUMAN DATA ARE EXPLAINED AS < absolute thickness of reticular basement membrane thickness and airway smooth muscle mass > WE DID NOT STAIN FOR COLLAGEN IN THE HUMAN BIOPSIES. THE IN VITRO DATA THEREFORE CANNOT BE COMPARED

Lines 430-433 this reads as a contradiction for Asp f13?  line 432-433 states that Asp f13 is less effective whereas l 431 indicates equivalent effects between WT and Asp F5 and f13 mutant strains? Please explain or adapt

THE RELEVANT SECTION CURRENTLY READS < Intranasal exposure of the mice to filtrates of wild type AF, and Asp f5 and Asp f13 null strains led to equivalent effects on Type 2 inflammation and airway responsiveness. The protease null strains, especially the Asp f 13 knockout, resulted in significantly reduced airway inflammation and remodeling than the wild type AF filtrate. AF may also be implicated in epithelial disruption. The serine protease Asp f13 and alkaline protease 1 (Alp 1), promote airway hyper-responsiveness by infiltrating the bronchial submucosa and disrupting the interactions between airway smooth muscle (ASM) and extracellular matrix (ECM) [82]. > I  HAVE MODIFIED THIS TO READ < Mice were nasally challenged with filtrates of wild type AF, and Asp f5 and Asp f13 null strains. The null strains had equivalent effects on Type 2 inflammation and airway responsiveness. The protease null strains, especially the Asp f 13 knockout, resulted in significantly reduced airway inflammation and remodeling compared with the wild type AF filtrate. AF may also be implicated in epithelial disruption. The serine protease Asp f13 and alkaline protease 1 (Alp 1), promote airway hyper-responsiveness by infiltrating the bronchial submucosa and disrupting the interactions between airway smooth muscle (ASM) and extracellular matrix (ECM) [82]. > I HOPE THIS CLARIFIES

Line 491 IL33 and its role, is this only in children? or also adults? Maybe the author can add such comment or explain the status

 INDEED, IN THE PARA CELL AND ANIMAL STUDIES IS THE SENTENCE < IL-33 has been implicated in both adult [37] and paediatric SAFS [5].> WHICH I HOPE ANSWERS THIS POINT

Textual

Lines

72/392/392/418/420 fungal names in italics; please check the complete manuscript

DONE, BUT IF I HAVE MISSED ANYTHING MAYBE THE COPYEDITOR CAN HELP ME

101-102/150-152  write names of all fungi used frequently as abbreviations or all in full length

DONE, BUT IF I HAVE MISSED ANYTHING MAYBE THE COPYEDITOR CAN HELP ME

132 rather CORRECTED

142 was 6% (add was) CORRECTED

147 start of the sentence is strange: remove such or write A similar study….CHANGED

150 determining serum specific IgE specific to fungal: remove underlined specific CORRECTED

175-176 font sizes differ CORRECTED

187 been admitted to the intensive care unit (ICU) admission and.. : remove underlined word CORRECTED

195 Dermatophagoides  emode : there is regularly some extra space between the words SORTED OUT

254 Rephrase : ..revealed that these children....SORTED

288 STRA and 371 ICS abbreviations were not yet introduced SORTED, THANKS

343 GWAS abbreviation needs introduction here; later in line 470 it is used in full SORTED, THANKS

438 to gain access to the ...SORTED

445/449 remodeling SORTED

452 reposition ref 74 SORTED

465: cytokines I HOPE SORTED – THE DOCUIMENT THE REVIEWER WORKED ON HAS DIFFERENT PAGE NUMBERS FROM THE ONE I HAVE DOWNLOADED

467/475  TH2 vs Th2 HOPEFULLY SORTED SHOULD BE TH2

541 so which in stead of ,so we what…FRUSTRATINGLY I CANNOT TRACE THIS GIVEN THE DIFFERENT PAGE NUMBERS, BUT HAPPY TO HAVE THE CHANGE MADE if I have missed it